# An analysis of forest biomass sampling strategies across scales

Jessica Hetzer[1], Andreas Huth[1, 2, 3], Thorsten Wiegand[1, 3], Hans Jürgen Dobner[4], Rico Fischer[1]

1 Department of Ecological Modelling, Helmholtz Centre for Environmental Research – UFZ, Leipzig, 04318, Germany
2 Institute of Environmental Systems Research, University of Osnabrück, Osnabrück, 49076, Germany
3 German Centre for Integrative Biodiversity Research (iDiv), Halle-Jena-Leipzig, 04103 Leipzig, Germany
4 Leipzig University of Applied Sciences- HTWK, Leipzig, 04277, Germany

*Correspondence to*: Jessica Hetzer (Jessica.hetzer@ufz.de)

**Abstract.** Tropical forests play an important role in the global carbon cycle as they store a large amount of carbon in their biomass. To estimate the mean biomass of a forested landscape, sample plots are often used, assuming that the biomass of these plots represents the biomass of the surrounding forest.

In this study, we investigated the conditions under which a limited number of sample plots conforms to this assumption. Therefore, minimum number of sample sizes for predicting the mean biomass of tropical forest landscapes were determined by combining statistical methods with simulations of sampling strategies. We examined forest biomass maps of Barro Colorado Island (50 ha), Panama (50,000 km$^2$), and South America, Africa and Southeast Asia (3 – 11 million km$^2$).

The results showed that around 100 plots (1-25 ha each) are necessary for continental-wide biomass estimations if the sampled plots are randomly distributed. However, locations of current inventory plots often do not meet this requirement, e.g. as their sampling design is based on spatial transects among climatic gradients. We show that these non-random locations lead to a much higher sampling intensity required (up to 54,000 plots for accurate biomass estimates for South America). The number of sample plots needed can be reduced using large distances (5 km) between the plots within transects.

We also applied novel point pattern reconstruction methods to account for aggregation of inventory plots in known forest plot networks. Results implied that current plot networks can have clustered structures that reduce the accuracy of large-scale estimates of forest biomass if no further statistical approach is applied. To establish more reliable biomass predictions across South American tropical forests, we recommend more spatially randomly distributed inventory plots (minimum 100 plots) and to ensure that the analyses of inventory plot data consider their spatial characteristics. The precision of forest attribute estimates depends on the sampling intensity and strategy.

## 1 Introduction

For a better understanding of the global carbon cycle, reliable estimations of aboveground biomass in vegetation have become increasingly important (Broich et al., 2009; Malhi et al., 2006; Marvin et al., 2014), especially for tropical forests, as they store more carbon in biomass than any other terrestrial ecosystem (Pan et al., 2011). Current biomass mapping approaches are based on forest field inventory plots (e.g., Chave et al., 2003; Lewis et al., 2004; Malhi et al., 2006; Mitchard et al., 2014) or remote

sensing measurements (e.g., Asner et al., 2013; Avitabile et al., 2016; Baccini et al., 2012; Saatchi et al., 2015) and involve statistical approaches (e.g., Malhi *et al.* (2006)) or vegetation modeling (e.g., Rödig *et al.* (2017)). Remote sensing-derived maps have a typical spatial resolution of 100-1000 m and capture the biomass of large landscapes or even entire continents (Asner et al., 2013; Avitabile et al., 2016; Baccini et al., 2012; Saatchi et al., 2011). In contrast, biomass maps based on field

inventories have a higher resolution so that the local distribution in biomass can be described in detail.

However, the biomass estimation of large forest landscapes by field inventory plots (typically between 0.25 and 1 ha) poses several challenges in the tropics. Firstly, field inventory campaigns of species-rich, densely grown tropical forests are costly and labor intensive, resulting in a much smaller number of available plots than in temperate and boreal regions (Schimel et al., 2015). Currently, tropical forests are sampled with less than one plot per 1000 km$^2$; a density that is up to 15 times less than

those that can be found in temperate zone (Schimel et al., 2015). For instance, the US national forest inventory includes more than 125,000 forest plots (Smith, 2002). This corresponds to 40 plots per 1000 km$^2$ In contrast, investigations of the South American Amazonian forest are often based on less than 500 forest plots (0.05 plots per 1000 km^2) (Lopez-Gonzalez et al., 2014; Mitchard et al., 2014) including a highly debated sampling error (Marvin et al., 2014; Mitchard et al., 2014; Saatchi et al., 2015).

Secondly, establishments of forest plots are often limited by accessibility e.g., due to topographic, logistic or political reasons (Houghton et al., 2009; Mitchard et al., 2014). Even if plots are representative for the landscapes (Anderson et al., 2009), extrapolations from clustered plot networks to larger scales can be biased (Fisher et al., 2008). Consequently, biomass estimations can include large uncertainties, e.g. like estimates of the total biomass of the Amazon (93 $\pm$ 23 PgC, based on 227 forest plots) include uncertainties of more than 25% (Malhi et al., 2006).

A first step to ensure reliable extrapolations of forest biomass from field plots to large scales is to determine how many plots would be necessary to accurately estimate mean biomass on a regional scale. Previous studies suggested that for regions of about 1,000 ha, 10-100 sampled one-hectare plots would be necessary (Marvin et al., 2014). However, most investigations assume that plots or biomass are distributed randomly in space (Chave et al., 2004; Fisher et al., 2008; Keller et al., 2015; Marvin et al., 2014) and therefore do not consider a possible bias due to the choice of sampling strategy. The selected sampling

design can significantly influence uncertainty and, consequently, the number of sample plots required (Clark and Kellner, 2012). A deeper understanding of how the choice of sampling design affects the number of plots required and the influence of the size of the plots is still missing.

In this study, we present a novel simulation approach for determining the number of plots necessary across scales answering the following questions: (I) How many sample plots are necessary for forest biomass estimations in South America and what

is the role of the sampling strategy? (II) What is the influence of scale on the sampling design?

More specifically, we analyze different sampling strategies for biomass in tropical forests at different scales: 50 ha (Barro Colorado Island), 50,000 km$^2$ (Panama, (Asner et al., 2013)) and 11 million km$^2$ (South America, (Baccini et al., 2012)). Following the scenario of a "virtual ecologist" (Zurell et al., 2010), we investigate through Monte Carlo simulations and

analytical investigations the plot size and sample size that are necessary for accurate biomass estimations. Furthermore, we
simulate nonrandom sampling strategies that imitate measurements of transects and real-world forest inventories.

## 2 Methods

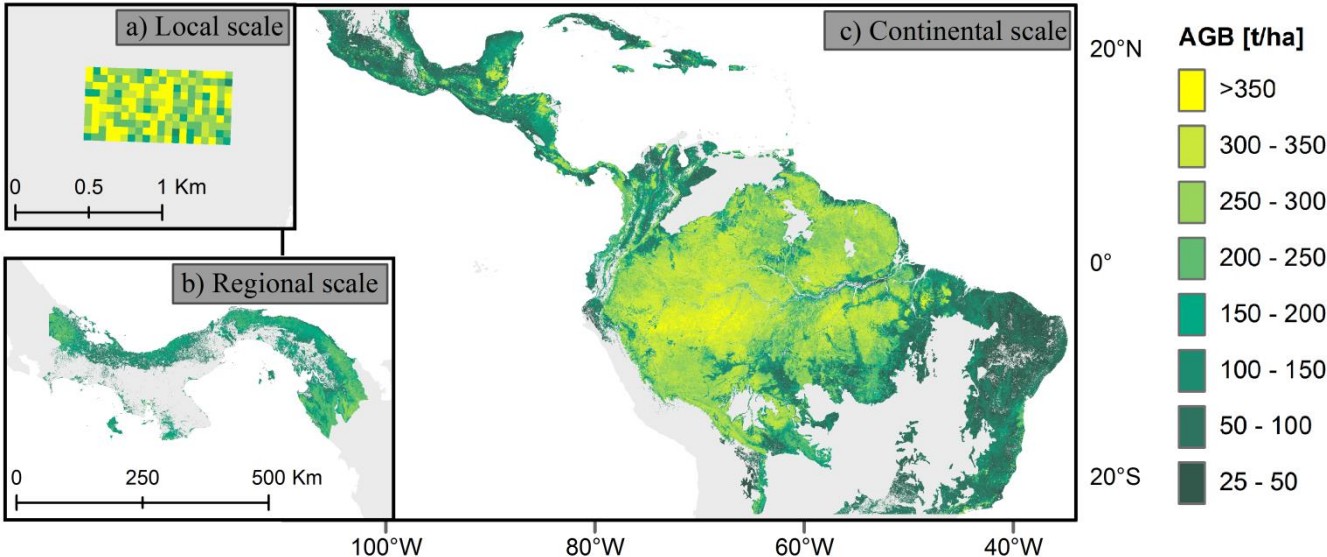

**Figure 1** Forest aboveground biomass (AGB) maps used for the study. **(a)** Biomass map of a forest plot on Barro Colorado Island (50 ha,
50 m resolution). **(b)** Biomass map for Panama (~50,000 km², 100 m resolution). **(c)** Biomass map for South America (~11 million km², 500
m resolution).For this study, we excluded all areas covering grasslands, savannas and shrublands**.**

### 2.1 Biomass maps at different scales

We focus on three forest biomass datasets for the South American tropical region covering different scales (Fig. 1). For an
analysis at the local scale, a biomass map of the Barro Colorado Island forest in Panama was applied (BCI, 50 ha) with
resolutions between 10 m and 100 m. The map was based on the forest inventory of 2010 (Condit et al., 2012), which included
measurements of all trees with a stem diameter greater than 1 cm (Condit, 1998). The aboveground biomass (AGB) per plot
was determined using allometric relationships (see Supplements of Knapp, Fischer and Huth (2018) for details).
Regional-scale analysis was carried out using a carbon density map of Panama that was derived from Airborne Light Detection
and Ranging (LiDAR) measurements from 2012, in combination with field measurements and satellite measurements (Asner
et al., 2013). The AGB values for this study were calculated by multiplying the carbon values by a factor of two. We aggregated
the AGB map from a 100 m resolution to resolutions of 200 m, 300 m, 400 m and 500 m. If aggregated pixels covered a
mixture of forest and non-forest areas, we assumed the non-forest areas to have a biomass of zero.

At the continental scale, we utilized a biomass map covering South America, Africa and South East Asia with a spatial resolution of 500 m (Baccini et al., 2012). Biomass values of this map give information on the aboveground vegetative biomass in the time period from 2008 to 2010 and were derived using a combination of MODIS data, LiDAR measurements and field data. For our analysis, we combined this biomass map with a biome map (Dinerstein et al., 2017) and excluded all areas that covered grasslands, savannas and shrublands as well as areas with an aboveground biomass of less than 25t/ha. In order to that, remaining areas could be assigned to one of the following four tropical and subtropical forest biomes: (a) Dry broadleaf forests (b) Moist broadleaf forests (c) Coniferous forests and (d) Mangroves.

Based on the continental forest biomass map of South America at a 500 m resolution, we constructed an additional biomass map of South America with a 100 m resolution using two different downscaling approaches (for details, see S3). The downscaling relationships were derived from the Panama map by upscaling this map from 100 m to 500 m resolution.

## 2.2 Simulated sampling strategies

We investigated three different sampling strategies: (a) random sampling (b) transect sampling and (c) clustered sampling (Fig. 2), with different sample sizes. For example, for analysis of the BCI forest, we divided the 50 ha biomass map into 200 square plots with a size of 50 m x 50 m (note that results could slightly differ if plots were circular (Lindsey et al., 1958)). Then, we ran simulations with sample sets containing one sample (0.25 ha), two samples (0.50 ha), and so forth until we reached a sample size of 100 samples (25 ha, half of the study area). For large-scale investigations, we analyzed sample sizes of up to 5,000 plots for Panama and 200,000 plots for South America.

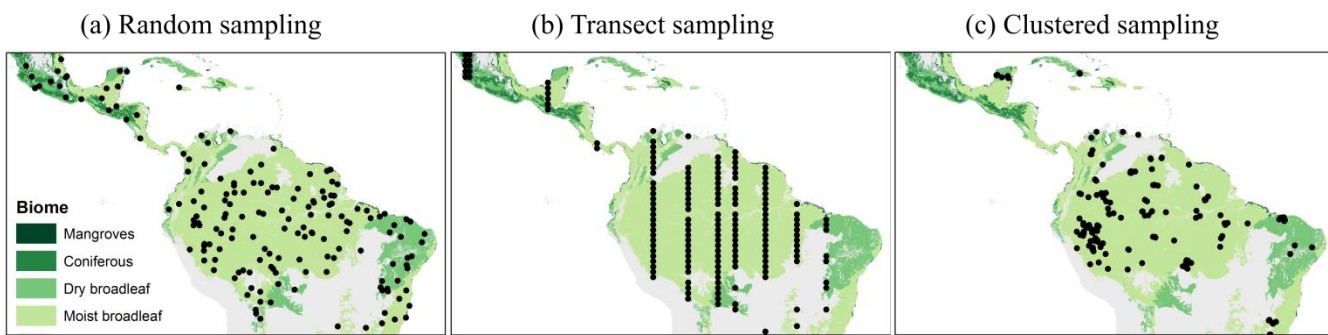

**Figure 2** Examples of different simulated sampling strategies for South America (colors indicate the tropical biome). Each black dot represents the location of one selected plot (25 ha). **(a)** Randomly distributed plots. **(b)** Transect samples (shown are strips with distances of 100 km between the plots). **(c)** Clustered samples (reconstructions of PP4).

## 2.2.1 Random sampling

Analysis of random sampling was performed using Monte Carlo simulations. For every map, we selected sampling plots at randomly selected positions (without replications) until the sample set reached the desired sample size. Random sampling is

the only strategy where we can assume that the spatial autocorrelation of the map does not influence analytical analysis using the central limit theorem (Supplement S1).

## 2.2.2 Transect sampling

Transect sampling mimics sampling strategies used whenever plots should cover different gradients (e.g., climate or soil gradients). In this case, field inventory plots are established in a straight line. In our simulation approach, we assume for simplification North-South transects that start at a randomly selected position of the map. Within one transect, the plots have regular distances of 0.5 km, 1 km or 5 km. Whenever the transect reaches the southern end of the map, a new randomly selected north-south transect is chosen starting at the northern border.

The analysis of Panama was conducted by selecting plots of 1 ha (map with 100 m resolution). For South America, we selected plots of 25 ha (map with 500 m resolution). To explore if the north-south climatic gradient influences the results, we also tested west-east instead of north-south tracks. However, the sampling performance remained similar (i.e., the probability of estimating the mean biomass accurately did not change considerably compared to north-south tracks).

## 2.2.3 Clustered sampling

The clustered sampling approach mimics the spatial clustering of real-world field inventory networks. To this end, we reconstructed the spatial pattern of the plot networks of four studies that estimated forest biomass, including Houghton *et al.* (2001), PP1; Poorter *et al.* (2015), PP2; Malhi *et al.* (2006), PP3; and Mitchard *et al.* (2014), PP4 and analyzed them separately regarding the South American map with a resolution of 500 m (25 ha plot size). After removing duplicate locations within the 500 m grid as well as plots that are located in grasslands, savannas or shrublands (according to Dinerstein et al., (2017)) the number of plots per network ranged between 23 and 167. To generate 1,000 plot networks with similar spatial configurations as the original ones, we applied the method of pattern reconstruction (Wiegand, He and Hubbell (2013); software "Pattern-Reconstruction"). This annealing method produces stochastic reconstructions of an observed point pattern that show the same spatial characteristics as the observed pattern, as quantified by several point pattern summary functions (for details see S2).

## 2.3 Determining the minimum sample size

For each map and each sample size $n$, we calculated the sampling probability $P_n$, which quantifies how often the mean of a sample equals the mean of the underlying "true" biomass distribution (under a given accuracy) as the relative frequency out of 1,000 sample sets. For each sample set, the mean biomass ($\overline{X_{i,n}}$ in t/ha) was estimated, where $i$ is the sample set number, and $n$ is the sample size. $\overline{X_{i,n}}$ was then compared with the "true" mean biomass, $\mu$ [t/ha], of the underlying biomass map. A sample set was assumed to be accurate if $\overline{X_{i,n}}$ was within $\mu \pm 10$ %. The sampling performance can be assessed as follows:

$$P_n \cong \frac{1}{1000} \sum_{i=1}^{1000} w_i, \text{ with } w_i = \begin{cases} 1, if \ \dfrac{|\overline{X_{i,n}} - \mu|}{\mu} \le 0.1 \\ 0, if \ \dfrac{|\overline{X_{i,n}} - \mu|}{\mu} > 0.1 \end{cases}$$

$P_n$ typically increases with the sample size from 0 (no sample could represent the mean biomass) to 1 (all samples could represent the biomass). We defined $n_{min}$ as the minimum sample size $n$, at which $P_n$ reaches 90 %. The minimum sampling area, $a_{min}$, is calculated by multiplying the number of plots, $n_{min}$, by the plot size.

## Results

### 3.1 Random sampling

#### 3.1.1 Local scale (Barro Colorado Island)

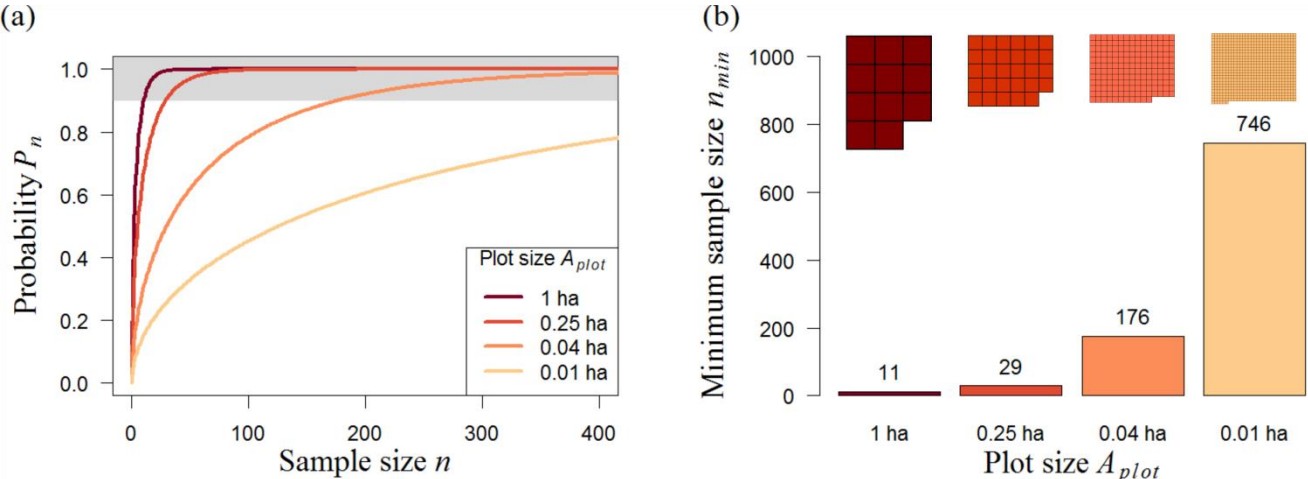

**Figure 3** Analysis of different random sampling strategies for the Barro Colorado Island forest (BCI, 50 ha). **(a)** Analytical results showing the number of plots and probability $P_n$ that the mean biomass of those plots reflects the mean biomass of the forest (for details, see Methods). We consider strategies using 0.01-1 ha plots (plot size, represented by line colors). The upper boundary (grey) marks sample sizes with at least 90% chance to meet the mean biomass of the original biomass map. **(b)** Necessary number of plots, $n_{min}$, to estimate the biomass reliably (minimum sample size from samples with $P_n \ge 90$ %). Shapes above the bars represent the necessary sampling area $a_{min} = A_{plot} \cdot n_{min}$.

The analysis of the 50 ha biomass map (BCI) show the expected result that samples with larger plot sizes produce more accurate biomass estimates (Fig. 3a). For instance, a randomly chosen 0.01-ha plot has a probability ($P_n$) of 5 % of representing the mean biomass of the whole BCI forest, but if the plot has an area of 1 ha, $P_n$ reaches 40 %. The size of the plots also affects the minimum number of plots required ($n_{min}$) for reliable biomass estimates (biomass estimates that have at least a 90% chance to meet the mean biomass of the original biomass map)**.** For small plots (plot size $\le$ 0.04 ha), $n_{min}$ decreases markedly (Fig. 3b). While only 11 one-hectare plots are needed to estimate the biomass, the number of plots increases to 176 if the plot size

is 0.04 ha (20 m x 20 m). However, the minimum total area of the samples $(a_{min})$ remains similar (Table 1, BCI); i.e., it makes no difference in sample performance whether the samples are taken from 29 plots of 0.25 ha each or 746 plots of 0.01 ha each, as an area of about 7 ha is sampled in both scenarios. Therefore, the most efficient sampling strategy for the 50 ha scale would involve 0.25 ha plots, as greater plot sizes would result in a greater total sampling area $(a_{min})$, and smaller plot sizes would simply increase the number of plots.

### 3.1.2 Regional scale (Panama)

Analyzing the biomass map of Panama (50,000 km$^2$) by using plot sizes between 1 ha and 25 ha (Table 1, Panama), we found that the minimum sample size ranges between 70 and 74 plots. In contrast to the BCI analysis, plot size has no remarkable influence on the minimum sample size. However, the total sampling area $(a_{min})$ increases from 70 ha to 1850 ha for different plot sizes. The most efficient sampling strategy at this scale is therefore to sample 70 plots of 1 ha each.

### 3.1.3 Continental scale (South America)

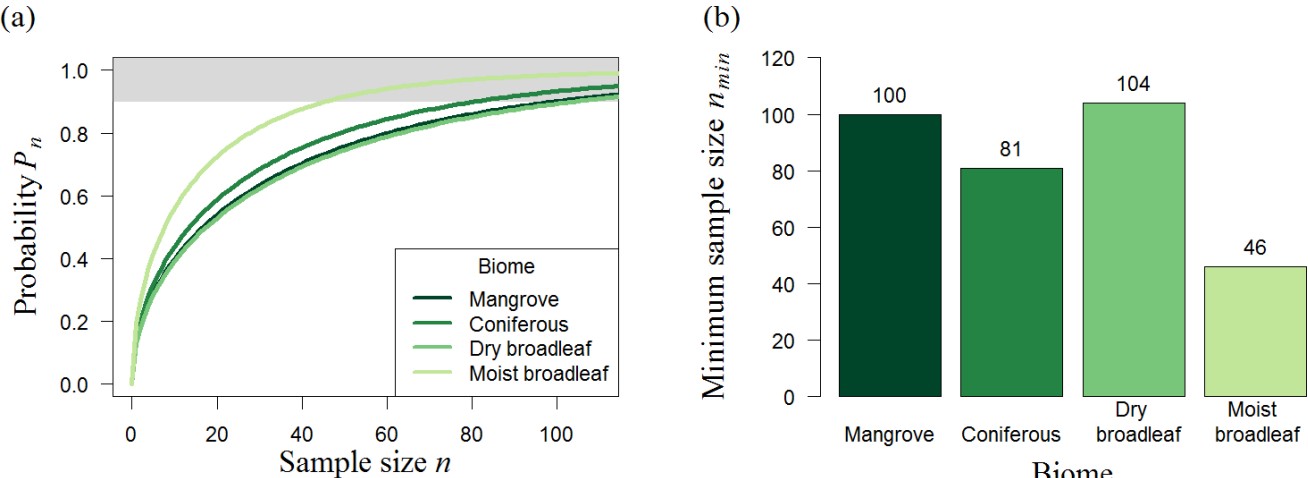

**Figure 4** Results of random sampling for different biomes of South America. (a) Analytical results showing the number of plots and probability $(P_n)$ that the mean biomass of those plots reflects the mean biomass of the forest biome (for details, see Methods). The upper boundary (grey) marks sample sizes with at least 90% chance to meet the mean biomass of the original biomass map. (b) Necessary number of 25 ha plots plots, $n_{min}$, to estimate the biomass for South America forest biomes reliably (minimum sample size from samples with $P_n \geq$ 90 %; displayed above the bars).

We found that the needed sampling number does not depend on the total forest area of biomes when samples are chosen randomly (Fig. 4). Mangrove forests (100 plots for 90,000 km$^2$) are the least abundant biome in South America but require a similar number of samples as dry broadleaf forest (104 plots for 2 Mio km$^2$). Furthermore, the minimum sample size does not increase compared to the Panama analysis(Table 1), e.g., plot number estimations for South America moist broadleaf forest (46 plots at 500m resolution) is even 35 % less than for the Panama forest (74 plots at 500m resolution; mainly consisting on moist broadleaf forest).

For the whole South America tropical forest (11 million km$^2$), 74 plots of 25 ha are necessary to estimate the mean biomass

180 with sufficient accuracy (Table 1, South America (500 m); for Africa and Southeast Asia, see Table S1). This corresponds to a total sampling area ($a_{min}$) of about 18.5 km$^2$.

Using the downscaling approach D1 (see Supplements S3 for details) we found that about 70 one-ha plots would be necessary to estimate the mean biomass of the South American tropical forest (Table 1, South America (100 m)). If we assume a much higher variation of biomass values than observed in the map (downscaling approach D2, see Supplements S3), this number

185 can rise to 121 one-ha plots.

**Table 1** Analyzed forest biomass maps and the corresponding minimum sample size. The forest biomass maps for South America (11,000,000 km$^2$, (Baccini et al., 2012; Dinerstein et al., 2017)), Panama (50,000 km$^2$,(Asner et al., 2013)) and Barro Colorado Island (50 ha, (Condit et al., 2012)) and their random sampling performance are shown. Different resolutions of the maps led to different results. The

190 minimum sample size refers to the necessary number of plots to accurately estimate the observed mean biomass of the forest (the mean of the samples does not deviate more than 10 % from the observed mean biomass with a probability of at least 90 %). The last column shows the necessary sampling area $a_{min} = A_{plot} \cdot n_{min}$.

| Map (Resolution) | Plot size $A_{plot}$ [ha] | Coefficient of variation CV [%] | Minimum sample size $n_{min}$ [plots] | Minimum total area of samples $a_{min}$ [ha] |
|---|---|---|---|---|
| South America (500 m) | 25 | 51.98 | 74 | 1850 |
| South America (100 m) | 1 | 50.63 | 70 | 70 |
| Panama (500 m) | 25 | 52.22 | 74 | 1850 |
| Panama (400 m) | 16 | 51.97 | 74 | 1184 |
| Panama (300 m) | 9 | 51.68 | 73 | 657 |
| Panama (200 m) | 4 | 51.27 | 72 | 288 |
| Panama (100 m) | 1 | 50.77 | 70 | 70 |
| BCI (100 m) | 1 | 19.32 | 11 | 11 |
| BCI (50 m) | 0.25 | 32.57 | 29 | 7.25 |
| BCI (20 m) | 0.04 | 80.55 | 176 | 7.04 |
| BCI (10 m) | 0.01 | 165.95 | 746 | 7.46 |

### 3.2 Transect sampling

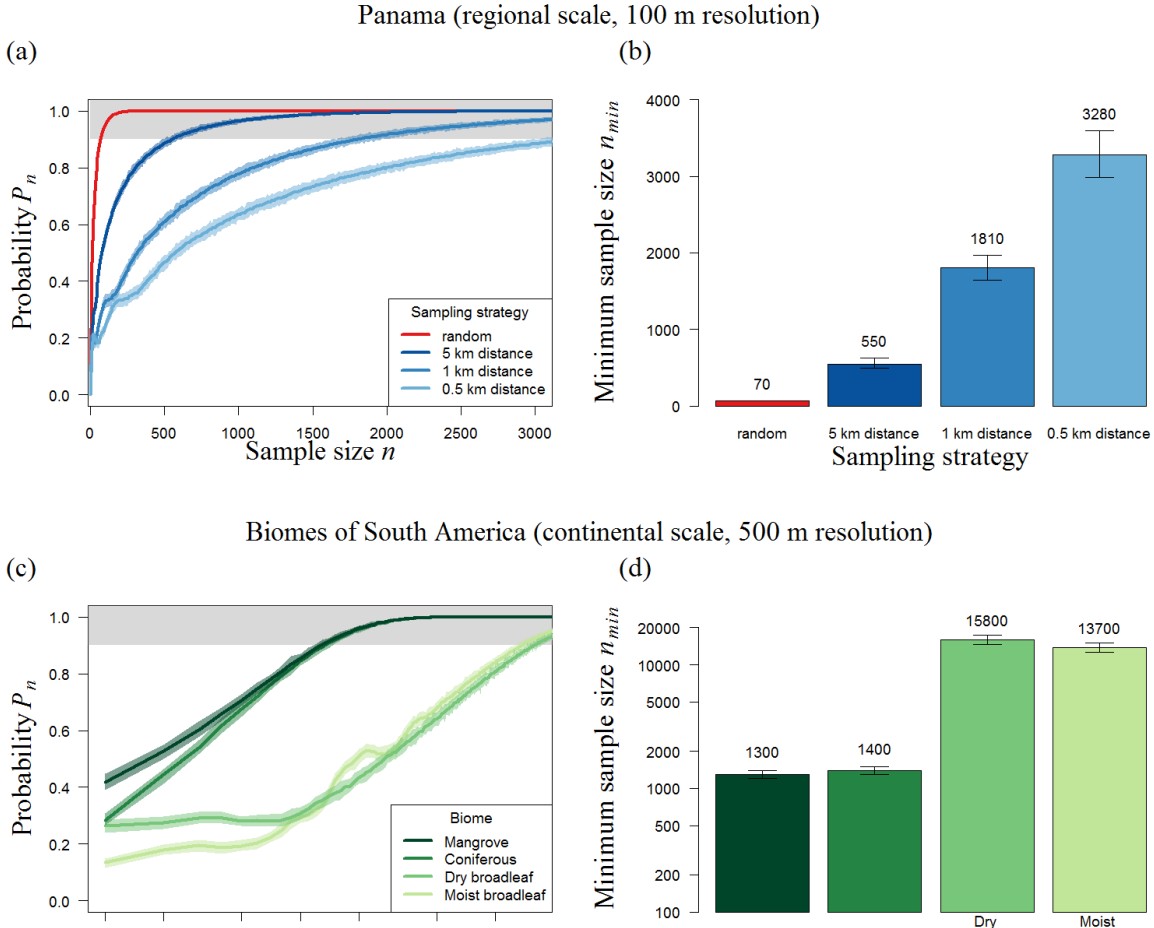

**Figure 5** Results of transect sampling for **(a-b)** Panama and **(c-d)** forest biomes in South America. **Left:** Simulation results showing the number of plots and probability ($P_n$) that the mean biomass of those plots reflects the mean biomass of the forest (for details, see Methods). **(a)** We focus on three strategies using distances of 500 m, 1 km and 5 km between plots (shown in blue) and compare them to random sampling (red). **(b)** Results for different biomes of the American tropical forest using a distance of 1 km. The area around each line indicates the 95 % confidence intervals of 100 repetitions (total of 1000*100 runs for each sample size). The upper boundary (grey) marks sample sizes with at least 90% chance to meet the mean biomass of the original biomass map. **Right:** Necessary number ($n_{min}$) of 1 ha plots for Panama and of 25 ha plots for biomes of South America (error bars show the 95 % confidence intervals of 100 repetitions).

The performance of non-random strategies was related to the spatial characteristics of maps (S4, Fig. S3). When the spatial clustering of the BCI forest biomass map is analyzed at the scale of 50 m, the obtained spatial biomass distribution is comparable to a random configuration; thus, the design of the sampling strategy has no influence on the results for this local

forest area. For Panama and South America, the biomass is distributed in such a way that similar biomass values are more likely to be close to each other, which leads to biased estimation of the mean biomass if the samples are close to each other (e.g., transects with distances of 0.5 km between the plots). This results in differences between random sampling and transect sampling (Fig. 5a): compared to random sampling, transect samples show a lower probability ($P_n$) of estimating the mean biomass of the forest accurately independent of the sample size. For Panama, random samples based on 100 one ha plots exhibit a $P_n = 95$ %, while transect samples are less than 60 % reliable (Fig. 5a).

The results show that if the distances between the plots increase from 0.5 km to 5 km, about 80 % fewer plots are necessary for accurate estimations. Larger distances between measurements within one transect make the strategy "more random", and it therefore performs better. Using distances of 5 km Panama requires a total sampling area of 550 ha (instead of 70 ha with random sampling) to estimate the biomass of the 50,000 km$^2$ forest with a sufficient precision (Fig. 5b). In summary, even with large distances between plots, transect sampling leads to higher sampling efforts than random sampling.

For South America (11 million km$^2$), transect sampling based on 100 plots (25 ha plot size) show a probability $P_n$ of less than 40 % (Fig. S4). Using distances of 5 km, the minimum sampling size increases by a factor of 140 compared to random sampling (Fig. S4), leading to a total sample area of about 2,500 km$^2$.

A stratification into forest biomes does not lead to a marked reduction of the overall number of needed sample plots, since the sum of the plots needed for all single biomes (in total 32,200; Fig. 5d) is similar to plots needed for an overall forest sampling (36,000 plots; Fig. S5). However in contrast to the random sampling, the area size of each biome affects the sampling effort. Here, the large broadleaf biomes need about ten times more transect samples than coniferous or mangrove forests.

## 3.3 Clustered sampling

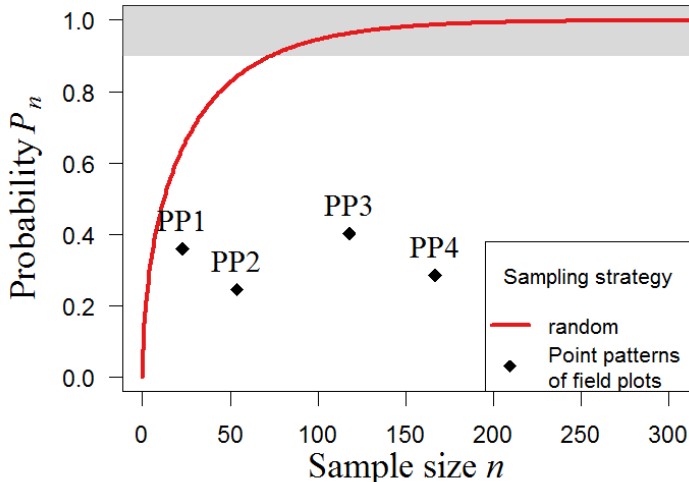

**Figure 6** Clustered sampling of biomass in South America. We tested different clustered sampling strategies using reconstructed point patterns based on the locations of existing field plots in South America (PP1-PP4). The simulation was performed with the South America map with a resolution of 500 m (25 ha plot size). Results show the probability ($P_n$) of accurate sampling for the spatial clustering of each point pattern (black crosses, accurate means that the mean biomass of the sample does not derive more than 10% from the mean biomass of the original map). The upper boundary (grey) marks sample sizes with at least 90% chance to estimate biomass accurately. As a reference, the results for random sampling are shown (red line).

Samples based on forest inventory plots are often influenced by accessibility, which leads to nonrandom locations of the sample plots that are simulated under the clustered sampling approach. Here, we examine the biomass map of South America with reconstructed point patterns PP1-PP4 based on the locations of existing inventories in South America (23-167 plots, see Methods). The results show that the probability ($P_n$) of estimating forest biomass accurately is considerably lower compared to the probability associated with random samples (Fig. 6). All samples present less than a 45 % chance of reflecting the real mean biomass for South America. For clustered sampling, a greater number of samples per se does not lead to better biomass estimations. The positions of the plots therefore play a crucial role. Although PP4 combines many plots of PP1 and PP3, the stochastic sampling scheme based on the spatial aggregation of plots cannot capture the biomass distribution substantially better than those based on the single datasets alone. In summary, the simulation results demonstrate that non-random strategies such as transect sampling and clustered sampling differ considerably from random sampling, leading to increased sampling efforts and noticeably greater sampling uncertainties.

## 4 Discussion

Due to the large area of tropical forest, only a few parts of the forest can be investigated in detail. Therefore, effective sampling strategies for these forests are relevant (Broich et al., 2009; Chave et al., 2004; Malhi et al., 2006; Marvin et al., 2014). The question of how many forest plots are necessary to predict forest biomass has not yet been fully answered. Thus far, sampling quality has often been determined on the basis of the assumption that samples are spatially randomly distributed (Chave et al., 2004; Fisher et al., 2008; Keller et al., 2015; Marvin et al., 2014). However, sampling at large scales in the tropics often does not fulfill this condition because in many cases, random locations are difficult to access (Wang et al., 2012). In this study, we compared different sampling strategies for tropical forests across various scales and plot sizes examining the probability to obtain the correct biomass estimate and the associated minimum sample size. Therefore, we analyzed random samples and compared them to simulated samples that are spatially clustered. Please note that in this study, we did not consider additional error sources, e.g., due to tree size measurements or allometric models, even though they are also known to influence biomass estimates (Chave et al., 2004).

### 4.1 Random sampling

Focusing on forests in South America we showed that independent from forest area, less than 100 randomly distributed one-ha plots are necessary to estimate the mean biomass with sufficient precision. This result is in line with a study by Marvin *et al.* (2014) predicting minimum sample sizes between 10 and 100 plots for forest regions in Peru (1-10 km$^2$).

By testing plot sizes between 0.01 ha and 1 ha, we demonstrated that inventory plots should not be smaller than 0.25 ha because smaller plots tend to be considerable more heterogeneous (reflected by a large increase of the CV) and lead to a noticeably greater number of necessary sample plots. Although the coefficient of variation (CV) of the biomass distribution increases with decreasing plot size for local forests (Réjou-Méchain et al., 2014; Wagner et al., 2010), there seem to be only small effects for larger landscapes. For Panama, we even found that biomass distribution of the aggregated maps were more heterogeneous due to averaging forest with non-forest areas. To estimate the minimum sample size of a particular forest region, it might be useful to explore biomass variability, for example, by using forest models (Zurell et al., 2010) or topography (Réjou-Méchain et al., 2014).

For large areas (tropical forests in South America, Africa, Southeast Asia), we obtained minimum sample sizes of 74-103 plots (randomly distributed, 25 ha each) on each continent. We also tested larger plot sizes with a biomass map from Saatchi *et al.* (2011), but the results were similar (75-136 plots, 100 ha each). Furthermore, we also tested smaller plot sizes by downscaling the South American biomass map to 100 m using relationships derived from the Panama forest biomass map (50,000 km$^2$ forest area). The analysis indicated that 70 plots of 1 ha that are randomly distributed in space are sufficient for biomass estimations in South America at large scales (Table 1).

Some remote sensing based biomass maps can miss empirically measured biomass patterns in tropical forests (Mitchard et al., 2014). Biomass maps might not meet the high variability of biomass in forests because they do not include fine scale variation

(Mitchard et al., 2014) and saturate at high biomass values (Lu, 2006; Sellers, 1985). We addressed this issue of missing fine scale variation by constructing an additional biomass map at 100 m resolution with much higher variation in biomass values than observed in the used biomass map. In this case, the number of one-ha plots that are necessary for continental estimates of the South American tropical forest increased to 121 one-ha plots (instead of 70 plots). Please note that we tested a simple downscaling procedure, so caution must be applied to these initial findings. In summary, we found that a higher variation in biomass values leads to a higher sampling effort (see figure S1).

## 4.2 Nonrandom sampling

Our analysis showed that sampling efforts change considerably if samples are not random in space. For South America, non-random samples of forests are less reliable and require substantially more plots to achieve accurate biomass estimations. This means that the necessary number of plots for non-random sampling strategies (as can be found in real-world inventories) cannot be assessed by Monte Carlo simulations that implicitly assume that samples are random (as in related studies of (Chave et al., 2004; Fisher et al., 2008; Keller et al., 2015; Marvin et al., 2014). Instead simulation procedures need to incorporate more advanced methods that include aggregated plot placement.

We demonstrated that a spatial autocorrelation has an effect on the sampling strategy (Legendre and Fortin, 1989; Réjou-Méchain et al., 2014) if plots close to each other are more similar than plots located farther apart (positive autocorrelation). Result suggest that for larger regions biomass tend to be more spatially clustered (e.g., large forest biomass occur more frequently within the Amazon basin than in the surrounding landscape) because biomass varies due to environmental gradients and geographical reasons (Houghton et al., 2009). Therefore, the uncertainty of large-scale estimations might be more affected by the sampling design than estimates for local scales. However, also small forested regions can be spatially correlated in terms of biomass (e.g., due to management or topography) so biases can't be excluded whenever the sampling design is not random. The sampling performance of current plot networks relates to an interplay of clustering and scattering of the inventory plots in certain areas. For instance, if inventory plots are more likely to be located in densely overgrown mature forests, the biomass is overrated because the rest of the plots cannot compensate this bias ("majestic forest bias" (Malhi et al., 2002)). We see a high potential in point pattern analysis to determine critical levels of aggregation that can bias sampling estimation. Current plot networks might improve their sampling performance by combining subsamples of aggregated plot clusters while including additional plots in poorly sampled regions.

Existing plot networks may provide better estimates than suggested by a "blind" sampling using additional information (e.g. climate and soil covariates). Those covariates can be utilized e.g., to define weighting factors that enhance biomass mean estimation. To analyze this issue, the pattern reconstruction approach used in this study could include additional criteria (ideally also those used for selection of plots). If the covariates can be mapped in the entire study area, the pattern reconstruction approach can take into account the additional constraints and reject plot configurations that do not agree with these criteria.

## 5 Conclusions

In summary, our study shows that the accuracy of the biomass estimates derived from samples depends considerably on the sampling strategy. Inventories are highly relevant for studying forest structure and dynamics. For South America, we have shown that more spatially randomly distributed plots are beneficial for continental-wide biomass estimations. For a given sampled area, plot size should not fall below 0.25 ha, as the variability of biomass values will strongly increase (Chave et al., 2003; Clark et al., 2001; Keller et al., 2015; Réjou-Méchain et al., 2014), and tree-level measurement errors can dominate (Chave et al., 2004).

To establish forest plots randomly across South America is challenging. On the one hand, mature tropical forests have high tree densities (Crowther et al., 2015), so measurements are more labor intensive. On the other hand, random plot locations may lead to large distances between the plots (Wang et al., 2012) making them more difficult to access and also results in higher efforts and costs.

Some studies combine field inventories with remote sensing data to estimate the biomass of large regions (Asner et al., 2013; Baccini et al., 2012; Rödig et al., 2017; Saatchi et al., 2011) as remote sensing can sample forest regions in a short time (Houghton et al., 2009; Schimel et al., 2015). The here shown transect sampling could give also hints for remote sensing based on airplane campaigns flying in straight lines over forest transects (e.g., comparable to Asner et al., 2013). The methods presented can be applied to any spatially clustered sampling technique. The sampling design is very relevant not only for forest biomass estimations, but also in view of other forest attributes (e.g., production). This should be considered when establishing forest plot networks.

## Code and Data availability

Biomass data (BCI, Panama, Tropics) is available in the corresponding reference. The R code for sampling simulations is available upon request from the corresponding author.

## Author contribution

JH, RF, and AH conceptualized the research; JH prepared the data and ran analyses. TW supported point pattern analysis. HJD contributed to analytical solutions. JH, RF and AH prepared the first draft of the manuscript and all the co-authors contributed substantially to subsequent versions, including the final draft.

## Competing interests

The authors declare no competing interests.

## Acknowledgements

We thank Greg Asner for providing the biomass data for Panama and Sassan Saatchi and Alessandro Baccini for providing
biomass values for the tropics. We thank Volker Grimm for his helpful comments on the manuscript and Franziska Taubert
for her support.

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
