# Peer review of "An analysis of forest biomass sampling strategies across scales"

_Biogeosciences, 2019_

## Referee Comment (RC1) · Anonymous Referee #1 · 11 Sep 2019

The paper by Hetzer et al. aims at assessing the effect of sampling strategy for estimating tropical forest aboveground biomass at different spatial scales. While this topic is of importance, it has already been well covered in the litterature. However, the simulated approach developped here have some originality (e.g. the point pattern reconstruction) but, in my opinion, some rather surprising or context-dependent results are due to methodological artefacts as described below. These artefacts are rather difficult to overcome but they should be at the minimum discussed or acknowledged before consideration for publication.

**Major comments**

Globally, many statements (see my specific comments) are very basic and already well known in the litterature (e.g. many sentences in the conclusion section). The author

should refer more to previous works and concepts, including those developped for temperate forests where a huge research effort on sampling strategy has been done in the past.

Investigating the effect of spatial scales (local, regional and continental) on sampling strategy is very appealing. However, I am very skeptikal about the use of remote sensing products as reference data. Both Asner and Baccini used passive optical data to extrapolate AGB at large scale and these products are well known to saturate for large AGB (>100-200 t/ha) values leading to a strong underestimation of AGB variability. This effect is well illustrated by the Fig. S2 where the SD of AGB first increase with AGB and then decrease. Theoretically the SD of AGB should continuously increase with the mean AGB (this is why people generally use CV instead of SD for comparison purpose). Thus, the decrease of SD with AGB in Fig. S2 is simply an illustration of the saturation problem so that using these maps, or dowscaling them using such SD pattern, result in a strong underestimation of AGB variation in high biomass areas, which, in my opinion, bring a strong bias in the final results presented here. This is probably the reason why some results are very counter-intuitive, such that plot size does not matter at large scale or that a large number of large plots provide less accurate AGB estimate than a small number of small plots (Lines 157-159).

I had two problems with the simulation of RS sampling. First, RS was simulated as discrete measurements, may be to simulate satellite LiDAR measurements such as those produced by GLASS or GEDI, but there is no justification for that (most satellites produce continuous measurements). This is surprising given that the authors used continuous RS-based maps to validate such RS sampling strategy, which look like a bit skizophrenic. Second, I did not fully understand the methodology. I understood that measurements were simulated at different distance along simulated transects but I did not understand how and if the distance between transects varied or not. I am not even sure that the authors simultaneously simulated several transects as would typically be done by a satelitte. I would suggest to simulate a sampling design similar to the one

that was or is adopted by GLASS or GEDI to make this simulation more practical even if this is challenging due to the high resolution of LiDAR footprint ($\sim$70 and 20 m) and the abovementioned downscaling problem.

The sampling showed in Fig. 2 illustrates a major problem. Nobody sample at the same time dense humid and dry forests to depict a mean biomass. This is always practically done by forest type using a prior stratification design. The minimum, to have something comparable with the other scales (BCI and Panama) is to focus only on tropical dense humid forests. This may explain why an aggregated sampling design produce such huge errors given that it sample very different forests at the continental scale.

As illustrated in Fig. S3, and by previous studies conducted in BCI, the spatial distribution in AGB do not significantly differs from a random distribution. This explain why, for a given sampled area, using several small or few large plots little impacts your estimates. This should be better explained in the present paper by explicitly mentioning the effect of spatial aggregation on sampling design and by stating that your result would probably not hold at the same scale in many (!) other forests that show strong AGB aggregation patterns (which is the case of most forests). Note also that the central limit theorem only applies if observations are independents (i.e., in absence of significant spatial structure), such that this theorem is theoretically valid only for the BCI scale in your study.

The discussion section may discuss the realism of a random sampling design at the continental scale in Amazonia.

The conclusion section should highlight more the originality of the present work.

**Specific comments**

Line 27: space lacking: "important(Broich"

Line 29: Are those referenced all provided biomass maps?

Line 34: Please replace by "so that the local distribution in biomass". At least remove

"local regions", which is inappropriate.

Lines 34-35: This last sentence is very vague.

Line 45: This is an old reference, what about most recent works such as Baccini and Saatchi maps?

Line 49: Assume that plots or biomass are. . ..

Line 52: I don't see the logic here. First it is obvious that the representativness of a given number of plots is context-dependent and varies with the total area of interest and second the number of plots fall into the recommendation cited line 48 so that it does not illustrate that the number of plots varies according to the sampling design.

Figure 1: I would have personally not call the b panel a landscape scale but rather a regional scale. I know that the definition of scale strongly varies in the litterature but I can hardly imagine a landscape of more than 500 km.

Line 73: "determined using allometric relationship" is really vague, unless the methodology is fully described in the Knapp paper. If yes, please add (see Knapp. . .. For details).

Lines 77-78: The following sentence is useless and confusing (strange to refer to plots for RS maps, we usually use pixels instead): "For this purpose, between 4 and 25 plots from the original map were averaged."

Lines 79-80: This last sentence is useless.

Line 81-82: This is not true that the Baccini map mostly derived from LiDAR measurements. The global methodology used was to callibrate GLASS LiDAR footprints with field data and then to calibrate a MODIS product with the calibrated LiDAR measurements. Thus the final product mostly reflect MODIS data, that are very little sensitive to biomass and highly sensitive to cloud cover (e.g. the large area of lower biomass observed on the western coastal area of central Africa, compared to the central basin,

:(

is simply due to cloud cover).

Line 83: Please provide rounded numbers.

Line 86-87: Please replace plot by pixel.

Fig. 3 legend: "below the bar " should be replaced by "above the bar"

Lines 143-144: Very obvious and well-known result.

Lines 154-155: First sentence useless.

Lines 180-181: Please reformulate.

Lines 230-232: Very obvious.

Line 235: If forest types are known a better strategy would be to stratify the sampling by forest types.

Lines 255-256: As already shown and discussed by previous works.

Line 259: What is a regional scale here?

Line 267: For a given sampled area, plot size should not.. ...

Line 270-271: This is what is generally done, remote sensing almost always relies on field data. Please be more explicit.

---

## Referee Comment (RC2) · Anonymous Referee #2 · 7 Oct 2019

This manuscript assesses the ability of different sampling strategies to characterise the overall mean biomass of tropical forests. Although there have been previous studies looking at this, the multi-scale approach and the point-pattern simulation to replicate the spatial clustering of previous studies add novelty, and mean that there is enough new for this to be a useful contribution. There are a number of issues that need to be addressed, primarily through improved discussion.

**Major comments**

I agree with the major points raised by Ref1, and won't elaborate on them more here except to say that it would make more sense to me to restrict the allocation of sampling points to a single biome (i.e. moist tropical forests) and areas with forest cover (i.e. above a given threshold in the Baccini map) to more realistically reflect real sampling

efforts.

The analysis of clustered sampling strategies implies a very naïve analysis approach to get an overall mean – just taking an average across plots without considering their configuration. To what extent the performance of clustered plot networks at estimating the overall mean can be improved by analyses accounting of climate and soil covariates and/or spatial autocorrelation to account for this oversampling? I would assume that there would be considerable potential to remove the disruptive effect of non-random sampling, and instead move estimates to a point on the random sampling curve equivalent effective sample size of spatially random plots. Thus existing plot networks, with appropriate analysis, may provide much better estimates of continental mean biomass than implied by this study.

It is worth noting that the remote sensing maps used as reference have serious limitations (some pointed out by Ref1). Most importantly, they miss the effect of species composition on biomass, which is driven by wood density and leads to marked spatial patterns in Amazonia. This isn't so much of a problem for this study if the remote sensing reference maps are interpreted as providing realistic examples of large-scale spatial variation in biomass, rather than as real references. I do wonder if this means the large scale reference maps underestimate the extent of fine scale variation due to compositional differences across soil types (for example).

**Specific comment**

The barplots in Figures 3 and 4 could be misinterpreted as giving strong evidence that big plots are best, as they show the that the smaller plot size the more plots are needed. It would be good to also display the change in the area of sampling needed (as is done in the text and table), as that is more relevant to sampling effort.

---

## Referee Comment (RC3) · Anonymous Referee #3 · 29 Oct 2019

This manuscript details an interesting and novel approach to estimating forest biomass using a dispersed cluster of forest inventory plots. However, in a way this is a "big data" solution to a problem where the solution does not necessarily consider all the variables necessary to making appropriate and constrained biomass estimates. Forest inventory plots are often chosen based on a wide variety of information including forest, soils, hydrology, topography, climate, etc. and are often not randomly chosen. The classical approach of positioning plots in strategic, representative areas often provides constrained and informed estimates of biomass. That said, now that we have huge amounts of remotely sensed data we can apply "big data" approaches to test the extents and limits of many ecological methodologies. I think that is the advantage of this manuscript—that it explores this space and does so in an interesting and informative

way.

I am honestly torn on whether Amazonia is a perfect test or worst-case scenario fort his methodology. Thinking through this, diversity is incredibly high and there are subset forest types within Amazonia. . ..some that rely heavily on topography/climate such as cloud forests, while there are also dry forests, seasonally flooded forests, and also wet forests. But again, this wide variability may actually be a strength of this approach. I would like to see if compared to temperate forests regions in N. America and Europe as well as boreal areas across the higher latitudes. It would be an interesting comparison to see if those systems diverge wildly from Amazonia.

My specific comments follow, but I think this paper has a lot of potential to drive how we think through sampling and forest inventory methodology. I applaud the creativity of the researchers.

Numbers indicates line nos.

14-16: 25 ha is a lot of forest to inventory. I am already thinking of the sheer amount of folks I have to hire.

26: Define vegetation specifically as aboveground biomass

28: Qualify aboveground C

45: These units seem wrong. Also, this is an older citation and only one citation given for what you indicate is a widely varying range.

30-55: In general good content, but the case needs to be made why uncertainties in rain forests are potentially higher than other forests. That would add to this section.

Figure 1 : Maybe flip the scale.

Figure 3: What do you mean by accurate here?

* A point. . ...many inventory plots on the ground are circular, but I don't see specifically

(unless I have missed it) but are you using circular or square estimation here? There are some deep literature that may be consulted here about the differences

Lindsey, A. A., Barton Jr, J. D., & Miles, S. R. (1958). Field efficiencies of forest sampling methods. Ecology, 428-444.

228-232: These section could be revisited to ensure clarity in how the results are framed.

234 – What do you mean by sampling effort increase w/ smaller sample size?

---

## Author Comment (AC1) · 29 Nov 2019

The paper by Hetzer et al. aims at assessing the effect of sampling strategy for estimating tropical forest aboveground biomass at different spatial scales. While this topic is of importance, it has already been well covered in the litterature. However, the simulated approach developed here have some originality (e.g. the point pattern reconstruction) but, in my opinion, some rather surprising or context-dependent results are due to methodological artefacts as described below. These artefacts are rather difficult to overcome but they should be at the minimum discussed or acknowledged before consideration for publication.

Thank you for your very helpful comments. We will prepare a revision of our manuscript that will follow your recommendations closely. The main changes will be:

a)     Consideration of the biome for the biomass sampling

We conducted an additional analysis where sampling was carried out only in tropical forest biomes. This could reduce the number of plots required for continental biomass estimates. A further stratification into single biomes did not decrease this number significantly. We plan to add and discuss these findings in the revised manuscript.

b)     Renaming the sampling method

We propose to rename the 'remote sensing sampling' method to 'transect sampling' and consider the implications for remote sensing in the discussion section (e.g., airplane tracks from LiDAR campaigns).

c)     Impact of more variation in high biomass values

The current analysis leads to conservative estimations of necessary sample plots. We agree that the tested biomass maps have limitations due to saturation effects. We will conduct additional analyses with higher variation in high biomass values and discuss the results.

We have added our responses to your comments in blue following each comment.

**Major comments**

Globally, many statements (see my specific comments) are very basic and already well known in the litterature (e.g. many sentences in the conclusion section). The author C1 should refer more to previous works and concepts, including those developed for temperate forests where a huge research effort on sampling strategy has been done in the past.

Thank you for this helpful comment. We will improve our introduction by including a paragraph discussing some state of the art sampling methods for temperate forests (e.g., national forest inventories in Europe and North America), where systematic sampling designs were established and evaluated (e.g., Keith et al., (2009).

Furthermore, we will discuss why these sampling approaches are more difficult to establish in tropical regions. One main challenge is that tropical forests are often more dense than temperate forests (about twice as many trees per km^2 (Crowther et al., 2015)), so measurements are more labor intensive. Another issue is that access to tropical forest regions is often restricted due to

topographic, logistic or political reasons (Houghton et al., 2009; Mitchard et al., 2014). This limits a comprehensive sampling as applied for temperate forests.

Investigating the effect of spatial scales (local, regional and continental) on sampling strategy is very appealing. However, I am very skeptikal about the use of remote sensing products as reference data. Both Asner and Baccini used passive optical data to extrapolate AGB at large scale and these products are well known to saturate for large AGB (>100-200 t/ha) values leading to a strong underestimation of AGB variability. This effect is well illustrated by the Fig. S2 where the SD of AGB first increase with AGB and then decrease. Theoretically the SD of AGB should continuously increase with the mean AGB (this is why people generally use CV instead of SD for comparison purpose). Thus, the decrease of SD with AGB in Fig. S2 is simply an illustration of the saturation problem so that using these maps, or dowscaling them using such SD pattern, result in a strong underestimation of AGB variation in high biomass areas, which, in my opinion, bring a strong bias in the final results presented here. This is probably the reason why some results are very counter-intuitive, such that plot size does not matter at large scale or that a large number of large plots provide less accurate AGB estimate than a small number of small plots (Lines 157-159).

Thank you for that important remark. It is true that the biomass maps used in our study have their limitations. However, these maps represent currently the only possibility to analyze continent-wide biomass distributions. To overcome specific biomass map artifacts we analyzed different biomass maps (Baccini et al., (2012) and Saatchi et al., (2011); see supplements, table S1).

We agree that the saturation effect has not been considered yet. Therefore we analyzed a second downscaling method, where we assume, as an extreme case, a strongly increasing trend between mean aboveground biomass and its standard deviation (see attached figure 1). In this case we found that about 150 one-ha plots (instead of 88 one-ha plots derived by the currently used approach) would be necessary for mean biomass estimations of the South America tropical forest (see attached figure 2). Thus, saturation will increase the number of sampling plots needed. We plan to add and discuss these results in the revised manuscript.

I had two problems with the simulation of RS sampling. First, RS was simulated as discrete measurements, may be to simulate satellite LiDAR measurements such as those produced by GLASS or GEDI, but there is no justification for that (most satellites produce continuous measurements). This is surprising given that the authors used continuous RS-based maps to validate such RS sampling strategy, which look like a bit skizophrenic.

Thank you for this important comment. After some critical reflection, we decided to call this method 'transect sampling' as we focus mainly on the establishment of empirical forest plots with this sampling method. We will discuss the relevance of this sampling strategy also for remote sensing applications in the discussion section, as this transect sampling could be interpreted as proxy for airplane flight tracks from lidar campaigns.

Second, I did not fully understand the methodology. I understood that measurements were simulated at different distance along simulated transects but I did not understand how and if the distance between transects varied or not. I am not even sure that the authors simultaneously simulated several transects as would typically be done by a satelitte. I would suggest to simulate a

sampling design similar to the one C2 that was or is adopted by GLASS or GEDI to make this simulation more practical even if this is challenging due to the high resolution of LiDAR footprint (~70 and 20 m) and the abovementioned downscaling problem.

As mentioned above, this transect method sampled plots in North-South transects. Within one transect, the plots had regular distances of 0.5km, 1km or 5 km. The spacing between transects was not regular, but randomly chosen. For the South America forest, for example, this method needed typically about 100 randomly chosen North-South transects (considering 1 km distances). We will revise the method section to clarify this approach.

The sampling showed in Fig. 2 illustrates a major problem. Nobody sample at the same time dense humid and dry forests to depict a mean biomass. This is always practically done by forest type using a prior stratification design. The minimum, to have something comparable with the other scales (BCI and Panama) is to focus only on tropical dense humid forests. This may explain why an aggregated sampling design produce such huge errors given that it sample very different forests at the continental scale.

Thank you, this is a good point. In the revised manuscript we will combine the biomass map with a biome map (Dinerstein et al., 2017) to distinguish between different vegetation types. This gives us the possibility to analyze the sampling strategies not only for tropical forest (covering moist broadleaf, dry broadleaf, conifer and mangrove forest) but also for forests of different biomes separately. After merging these two maps, we found that the number of sampling plots decreases if taking only tropical forest into account (from 48,000 to 36,000 plots, see attached figure 2, first two bars). The additional analysis indicated furthermore, that a stratification into biomes does not lead to a significant reduction of the needed sample plots, since the sum of the plots needed for single biomes (34,000 plots) is similar to the plots needed for an overall forest sampling (36,000 plots). However, this stratification helps to better evaluate sampling effort for each biome. We will add the new results (see attached Figure 2) and discuss them in the revised manuscript.

As illustrated in Fig. S3, and by previous studies conducted in BCI, the spatial distribution in AGB do not significantly differs from a random distribution. This explain why, for a given sampled area, using several small or few large plots little impacts your estimates. This should be better explained in the present paper by explicitly mentioning the effect of spatial aggregation on sampling design and by stating that your result would probably not hold at the same scale in many (!) other forests that show strong AGB aggregation patterns (which is the case of most forests).

Thank you for your comment. We will revise the discussion by mentioning the effect of spatial aggregation (e.g., Chave et al., (2003); Marvin et al., (2014)).

Note also that the central limit theorem only applies if observations are independents (i.e., in absence of significant spatial structure), such that this theorem is theoretically valid only for the BCI scale in your study.

Thank you for that note, this is a tricky point. It is true that the large scale biomass maps show spatial autocorrelation, but we choose in our sampling strategy "random sampling" the biomass values from random locations of the map. This secures that each of our biomass observation is randomly generated in a way that does not depend on the values of the other biomass observations.

We would have an autocorrelation problem with the used biomass map if we apply for example clustered sampling (i.e., select always nearby points with higher probability). In our study we compared the outcome of the central limit theory only with the random sampling, not with the other sampling strategies. For all other sampling strategies the spatial autocorrelation of the underlying biomass map is crucial – therefore we simulated these sampling strategies instead of applying the central limit theorem. We will clarify this in the revised manuscript.

The discussion section may discuss the realism of a random sampling design at the continental scale in Amazonia.

We will add a paragraph on the realism of sampling in the discussion section (i.e., current amount of inventory plots in Amazonia, current remote sensing measurements).

The conclusion section should highlight more the originality of the present work.

We will revise the conclusion carefully regarding this comment. Highlights are for example novel methods to investigate non-random sampling strategies (e.g., by using point pattern simulations).

**Specific comments**

Line 27: space lacking: "important(Broich" Will be done.

Line 29: Are those referenced all provided biomass maps? In this study, we compared the biomass maps of Baccini et al., (2012) and Saatchi et al., (2011), but there are more biomass maps available (e.g.,Avitabile et al., (2016)). We will clarify this point.

Line 34: Please replace by "so that the local distribution in biomass". At least remove C3 "local regions", which is inapropriate. Will be done.

Lines 34-35: This last sentence is very vague. Will be deleted.

 Line 45: This is an old reference, what about most recent works such as Baccini and Saatchi maps? We will adjust the numbers to latest literature.

Line 49: Assume that plots or biomass are. . .. Will be done.

Line 52: I don't see the logic here. First it is obvious that the representativness of a given number of plots is context-dependent and varies with the total area of interest and second the number of plots fall into the recommendation cited line 48 so that it does not illustrate that the number of plots varies according to the sampling design. We agree that this number does not reflect the differences between sampling strategies. We will delete this example and revise the paragraph.

Figure 1: I would have personally not call the b panel a landscape scale but rather a regional scale. I know that the definition of scale strongly varies in the litterature but I can hardly imagine a landscape of more than 500 km. Thank you. We will rename the term.

Line 73: "determined using allometric relationship" is really vague, unless the methodology is fully described in the Knapp paper. If yes, please add (see Knapp. . .. For details). The methodology is described in the Knapp paper. We will add the reference as proposed.

Lines 77-78: The following sentence is useless and confusing (strange to refer to plots for RS maps, we usually use pixels instead): "For this purpose, between 4 and 25 plots from the original map were averaged." We will revise this sentence.

Lines 79-80: This last sentence is useless. Will be deleted.

Line 81-82: This is not true that the Baccini map mostly derived from LiDAR measurements. The global methodology used was to callibrate GLASS LiDAR footprints with field data and then to calibrate a MODIS product with the calibrated LiDAR measurements. Thus the final product mostly reflect MODIS data, that are very little sensitive to biomass and highly sensitive to cloud cover (e.g. the large area of lower biomass observed on the western coastal area of central Africa, compared to the central basin, C4 is simply due to cloud cover). Thank you for mentioning this important point. We will revise this sentence.

Line 83: Please provide rounded numbers. Will be done.

Line 86-87: Please replace plot by pixel. Will be done.

Fig. 3 legend: "below the bar " should be replaced by "above the bar" Will be done.

Lines 143-144: Very obvious and well-known result. We will add "Likewise to many other studies we show …".

Lines 154-155: First sentence useless. Will be deleted.

Lines 180-181: Please reformulate. Will be done.

Lines 230-232: Very obvious. We will revise this sentence.

Line 235: If forest types are known a better strategy would be to stratify the sampling by forest types. We agree with you and will show that by the expanded results on biomes.

Lines 255-256: As already shown and discussed by previous works. We will add references.

Line 259: What is a regional scale here? Forest sites comparable to BCI. We will revise this sentence.

Line 267: For a given sampled area, plot size should not. . … Will be done.

Line 270-271: This is what is generally done, remote sensing almost always relies on field data. Please be more explicit. We will revise our conclusions as we proposed above.

**Figures**

[Figure]

Figure 1: Comparison of downscaling approaches. **a)** Subplot heterogeneity in the Panama biomass map (500 m resolution). Shown is the aboveground biomass (AGB) at a 500 m resolution and the standard deviation (SD) of its associated 25 subplots at a 100 m resolution. Each dot represents one plot from the Panama map (~300,000 plots).The green line shows the 'mean value approach', as it was implemented in the current study. The pink line shows the proposed second downscaling approach. There, the linear trend resulting from AGB values smaller than 100 t/ha is continued for larger biomass values. With this second approach we assume a much higher variation in large biomass values as observed in the map. **b-c)** Aboveground biomass distribution of South America at a 100 m resolution using b) the mean value approach as implemented in the current study (green) and c) the linear trend approach (pink). Coefficient of variation (CV) and the minimum sample size ($n_{min}$) of randomly chosen 1 ha plots are displayed at the upper right corner for each biomass distribution.

[Figure]

Figure 2: Necessary number of samples to derive accurate mean estimations for different forest biomes of South America by applying the transect sampling (former remote sensing sampling). Samples (25 ha each) were taken with regular distances of 1 km between plots. The first bar shows the results for South America as implemented in the current study (Tropical vegetation). The second bar displays the number of plots when sampling is carried out exclusively in forest biomes. Therefore we merged the biomass map used (Baccini et al., 2012) with a biome map (Dinerstein et al., 2017), restricting sampling to moist broadleaf, dry broadleaf, coniferous and mangrove forest. The last four bars give the minimum sample size if forest biomes are sampled separately. Error bars reflect the range of 10 repetitions.

**Literature**

[revised manuscript text omitted]

---

## Author Comment (AC2) · 29 Nov 2019

This manuscript assesses the ability of different sampling strategies to characterize the overall mean biomass of tropical forests. Although there have been previous studies looking at this, the multi-scale approach and the point-pattern simulation to replicate the spatial clustering of previous studies add novelty, and mean that there is enough new for this to be a useful contribution. There are a number of issues that need to be addressed, primarily through improved discussion.

Thank you for your very helpful comments. We will prepare a revision of our manuscript that will follow your recommendations closely. The main changes will be:

a)      Restriction of sampling to forest biomes

Following your suggestion we analyzed the sampling strategies for each biome separately (covering moist broadleaf, dry broadleaf, conifer and mangrove forest). Results will be added and discussed in the revised manuscript.

b)      Discussion about the impact of more variation in high biomass values

The current analysis leads to more conservative estimations. We agree that the tested maps have limitations concerning the fine scale variation. Assuming an increased variation in biomass values would lead to a moderate increase in the minimum sample size. We will add this aspect in the discussion.

We have added our responses to your comments in blue following each comment.

**Major comments**

I agree with the major points raised by Ref1, and won't elaborate on them more here except to say that it would make more sense to me to restrict the allocation of sampling points to a single biome (i.e. moist tropical forests) and areas with forest cover (i.e. above a given threshold in the Baccini map) to more realistically reflect real sampling efforts.

Thank you for this comment. We will extend the study by analyzing the sampling strategies across different biomes. Therefore, the biomass map used for continental analyses (Baccini et al., 2012) is merged now with a global biome map (Dinerstein et al., 2017). To exclude rarely vegetated pixels within biomes, we assume a minimum above ground biomass threshold of 25 t/ha. Current results show that there are differences between biomes regarding the sampling effort (e.g., between moist broadleaf forests and conifer forests, see attached figure 1). We plan to include and discuss these additional results in the revised manuscript.

The analysis of clustered sampling strategies implies a very naïve analysis approach to get an overall mean – just taking an average across plots without considering their configuration. To what extent the performance of clustered plot networks at estimating the overall mean can be improved by analyses accounting of climate and soil covariates and/or spatial autocorrelation to account for this oversampling? I would assume that there would be considerable potential to remove the disruptive effect of non-random sampling, and instead move estimates to a point on the random sampling curve equivalent effective sample size of spatially random plots. Thus existing plot

networks, with appropriate analysis, may provide much better estimates of continental mean biomass than implied by this study.

This is a good point. We designed our analysis primarily to explore the effect of spatially clustered vs. random sampling. We therefore agree that existing plot networks, that stratified plots based on additional constraints, may provide better estimates than suggested by a "blind" clustered sampling.

A possible solution to this issue is to expand the pattern reconstruction approach to include additional criteria (ideally those used for selection of the real clustered plot networks, accounting e.g., for climate and soil covariates). If the covariates representing the additional criteria can be mapped in the entire study area, the pattern reconstruction approach can take the additional constraints into account and reject plot configurations that do not agree with these criteria.

However, we believe that such an analysis would be beyond the scope of our current study, but an interesting task for forthcoming studies. We therefore will briefly discuss in the discussion section that our clustered sampling strategies do not account for additional criteria that will be used for the design of real plot networks, and propose the above solution for a better assessment of the performance of clustered plot networks.

It is worth noting that the remote sensing maps used as reference have serious limitations (some pointed out by Ref1). Most importantly, they miss the effect of species composition on biomass, which is driven by wood density and leads to marked spatial patterns in Amazonia. This isn't so much of a problem for this study if the remote sensing reference maps are interpreted as providing realistic examples of large-scale spatial variation in biomass, rather than as real references. I do wonder if this means the large scale reference maps underestimate the extent of fine scale variation due to compositional differences across soil types (for example).

Thank you for your comment. We agree that continental biomass maps have their limitations especially in terms of fine scale variation. An higher variation of the biomass variation will lead to a higher sampling effort, such that our estimated plot number could be interpreted as a conservative estimation. We plan to add an additional analysis where we assume higher variations in high biomass values (see figure 1 in the response to the first Referee) and will discuss this important issue in the revised manuscript.

**Specific comment**

The barplots in Figures 3 and 4 could be misinterpreted as giving strong evidence that big plots are best, as they show the that the smaller plot size the more plots are needed. It would be good to also display the change in the area of sampling needed (as is done in the text and table), as that is more relevant to sampling effort.

Thank you. We will revise these figures and will show than also total sampling area.

**Figures**

[Figure]

Figure 1: Necessary number of samples to derive accurate mean estimations for different forest biomes of South America by applying the transect sampling (former remote sensing sampling). Samples (25 ha each) were taken with regular distances of 1 km between plots. The first bar shows the results for South America as implemented in the current study (Tropical vegetation). The second bar displays the number of plots when sampling is carried out exclusively in forest biomes. Therefore we merged the biomass map used (Baccini et al., 2012) with a biome map (Dinerstein et al., 2017), restricting sampling to moist broadleaf, dry broadleaf, coniferous and mangrove forest. The last four bars give the minimum sample size if forest biomes are sampled separately. Error bars reflect the range of 10 repetitions.

**Literature**

Baccini, A., Goetz, S. J., Walker, W. S., Laporte, N. T., Sun, M., Sulla-Menashe, D., Hackler, J., Beck, P. S. A., Dubayah, R., Friedl, M. A., Samanta, S. and Houghton, R. A.: Estimated carbon dioxide emissions from tropical deforestation improved by carbon-density maps, Nat. Clim. Chang., 2(3), 182–185 [online] Available from: http://dx.doi.org/10.1038/nclimate1354, 2012.

Dinerstein, E., Olson, D., Joshi, A., Vynne, C., Burgess, N. D., Wikramanayake, E., Hahn, N., Palminteri, S., Hedao, P., Noss, R., Hansen, M., Locke, H., Ellis, E. C., Jones, B., Barber, C. V., Hayes, R., Kormos, C., Martin, V., Crist, E., Sechrest, W. E. S., Price, L., Baillie, J. E. M., Weeden, D. O. N., Suckling, K., Davis, C., Sizer, N., Moore, R., Thau, D., Birch, T., Potapov, P., Turubanova, S., Tyukavina, A., Souza, N. D. E., Pintea, L., Brito, J. C., Llewellyn, O. A., Miller, A. G., Patzelt, A., Ghazanfar, S. A., Timberlake, J., Klöser, H., Shennan-farpón, Y. and Kindt, R.: An Ecoregion-Based Approach to Protecting Half the Terrestrial Realm, Bioscience, 67(6), doi:10.1093/biosci/bix014, 2017.

---

## Author Comment (AC3) · 29 Nov 2019

This manuscript details an interesting and novel approach to estimating forest biomass using a dispersed cluster of forest inventory plots. However, in a way this is a "big data" solution to a problem where the solution does not necessarily consider all the variables necessary to making appropriate and constrained biomass estimates. Forest inventory plots are often chosen based on a wide variety of information including forest, soils, hydrology, topography, climate, etc. and are often not randomly chosen. The classical approach of positioning plots in strategic, representative areas often provides constrained and informed estimates of biomass. That said, now that we have huge amounts of remotely sensed data we can apply "big data" approaches to test the extents and limits of many ecological methodologies. I think that is the advantage of this manuscript that it explores this space and does so in an interesting and informative way.

Thank your helpful comments. We will prepare a revision of our manuscript that will follow your recommendations. The main changes will be:

a)      Current forest sampling strategies

        We will add paragraphs on current sampling strategies in temperate forests to the introduction. Furthermore, we will discuss the feasibility of adapting those strategies in tropical forests.

We have added our responses to your comments in blue following each comment.

I am honestly torn on whether Amazonia is a perfect test or worst-case scenario fort his methodology. Thinking through this, diversity is incredibly high and there are subset forest types within Amazonia. . ..some that rely heavily on topography/climate such as cloud forests, while there are also dry forests, seasonally flooded forests, and also wet forests. But again, this wide variability may actually be a strength of this approach.

We investigated also the sampling strategies for other continents (see Supplements, Table S1). Results showed that the needed number of randomly sampled plots are higher for the tropical forests of South East Asia (131 plots) and Africa (185 plots) indicating that the tropical forests of Amazonia (102 plots) might be not the worst case, but the best-case-scenario.

I would like to see if compared to temperate forests regions in N. America and Europe as well as boreal areas across the higher latitudes. It would be an interesting comparison to see if those systems diverge wildly from Amazonia.

Thank you for raising this point. Comparisons with temperate regions would be very interesting even though we believe that this would go beyond the scope of this study. For higher latitudes we expect differences in the overall biomass distribution (e.g. due to a different species pool compared to the tropics) but also in the spatial distribution of biomass (e.g., due to forest management). Therefore, there might be also relevant differences in the amount of forest sampling plots needed.

However, forests in temperate regions are mostly monitored with already sophisticated sampling methods (i.e., national forest inventories in North America and Europe).  In the revised manuscript,

we will add some text about state of the art sampling methods for temperate forests and discuss why these sampling approaches are more difficult to establish in tropical regions (e.g., limited accessibility).

My specific comments follow, but I think this paper has a lot of potential to drive how we think through sampling and forest inventory methodology. I applaud the creativity of the researchers.

Thank you very much. We appreciate.

Numbers indicates line nos.

14-16: 25 ha is a lot of forest to inventory. I am already thinking of the sheer amount of folks I have to hire. We agree.

26: Define vegetation specifically as aboveground biomass Will be done.

28: Qualify aboveground Will be done.

45: These units seem wrong. Also, this is an older citation and only one citation given for what you indicate is a widely varying range. We will revise this statement.

30-55: In general good content, but the case needs to be made why uncertainties in rain forests are potentially higher than other forests. That would add to this section. We will add a section were we compare sampling strategies of tropical forests with those from temperate forest as mentioned above.

Figure 1 : Maybe flip the scale. Will be done

Figure 3: What do you mean by accurate here? With accurate we mean those samples where the mean biomass of the sample does not differ more than 10 % from the mean biomass of the original biomass map. We will revise this sentence to clarify the term.

* A point. . ..many inventory plots on the ground are circular, but I don't see specifically (unless I have missed it) but are you using circular or square estimation here? There are some deep literature that may be consulted here about the differences Lindsey, A. A., Barton Jr, J. D., & Miles, S. R. (1958). Field efficiencies of forest sampling methods. Ecology, 428-444.

Thank you for this comment. We are using square plots as we are limited by the spatial resolution of the biomass map. We will add this point to the methods. Nevertheless, the consequences of square vs. circular plots for the sampling effort is very interesting and we will add findings derived by Lindsey et al. to the introduction.

228-232: These section could be revisited to ensure clarity in how the results are framed. Thank you for this comment. We will improve this section by rephrasing several sentences.

234 – What do you mean by sampling effort increase w/ smaller sample size? We will delete this sentence and replaced it.

---

## Author Response (AR1)

The paper by Hetzer et al. aims at assessing the effect of sampling strategy for estimating tropical forest aboveground biomass at different spatial scales. While this topic is of importance, it has already been well covered in the litterature. However, the simulated approach developed here have some originality (e.g. the point pattern reconstruction) but, in my opinion, some rather surprising or context-dependent results are due to methodological artefacts as described below. These artefacts are rather difficult to overcome but they should be at the minimum discussed or acknowledged before consideration for publication.

Thank you for your very helpful comments. We prepared a revision of our manuscript that follows your recommendations closely. The main changes are:

a)      Consideration of the biome for the biomass sampling

We conducted additional analyses where sampling was carried out only in tropical forest biomes. This could reduce the number of plots required for continental biomass estimates. A further stratification into single biomes did not decrease this number significantly. We added and discuss these findings in the revised manuscript.

b)      Renaming the sampling method

We renamed the 'remote sensing sampling' method to 'transect sampling' and considered the implications for remote sensing in the conclusions section (e.g., airplane tracks from LiDAR campaigns).

c)      Impact of more variation in high biomass values

The current analysis leads to conservative estimations of necessary sample plots. We agree that the tested biomass maps have limitations due to saturation effects. We conducted an additional analysis with higher variation in high biomass values and discuss the results in the revised manuscript.

We have added our responses to your comments in blue following each comment.

**Major comments**

Globally, many statements (see my specific comments) are very basic and already well known in the litterature (e.g. many sentences in the conclusion section). The author C1 should refer more to previous works and concepts, including those developed for temperate forests where a huge research effort on sampling strategy has been done in the past.

Thank you for this helpful comment. We improved our introduction by including a paragraph discussing some state of the art sampling methods for temperate forests where systematic sampling designs were established and evaluated (lines 39-44).

Furthermore, we discuss why these sampling approaches are more difficult to establish in tropical regions. One main challenge is that tropical forests are often more dense than temperate forests (about twice as many trees per km^2 (Crowther et al., 2015)), so measurements are more labor intensive. Another issue is that access to tropical forest regions is often restricted due to

topographic, logistic or political reasons (Houghton et al., 2009; Mitchard et al., 2014). This limits a comprehensive sampling as applied for temperate forests. (lines 312-315).

Investigating the effect of spatial scales (local, regional and continental) on sampling strategy is very appealing. However, I am very skeptikal about the use of remote sensing products as reference data. Both Asner and Baccini used passive optical data to extrapolate AGB at large scale and these products are well known to saturate for large AGB (>100-200 t/ha) values leading to a strong underestimation of AGB variability. This effect is well illustrated by the Fig. S2 where the SD of AGB first increase with AGB and then decrease. Theoretically the SD of AGB should continuously increase with the mean AGB (this is why people generally use CV instead of SD for comparison purpose). Thus, the decrease of SD with AGB in Fig. S2 is simply an illustration of the saturation problem so that using these maps, or dowscaling them using such SD pattern, result in a strong underestimation of AGB variation in high biomass areas, which, in my opinion, bring a strong bias in the final results presented here. This is probably the reason why some results are very counter-intuitive, such that plot size does not matter at large scale or that a large number of large plots provide less accurate AGB estimate than a small number of small plots (Lines 157-159).

Thank you for that important remark. It is true that the biomass maps used in our study have their limitations. However, these maps represent currently the only possibility to analyze continent-wide biomass distributions. To overcome specific biomass map artifacts we analyzed different biomass maps (Baccini et al., (2012) and Saatchi et al., (2011); see supplements, table S1).

We agree that the saturation effect has not been considered yet. Therefore we analyzed a second downscaling method, where we assume, as an extreme case, a strongly increasing trend between mean aboveground biomass and its standard deviation (see revised Supplements, section S3). In this case we found that about 121 one-ha plots (instead of 70 one-ha plots derived by the currently used approach) would be necessary for mean biomass estimations of the South America tropical forest.  Thus, saturation will increase the number of sampling plots needed. We added (lines 183-186) and discussed (lines 278-283) these results in the revised manuscript.

I had two problems with the simulation of RS sampling. First, RS was simulated as discrete measurements, may be to simulate satellite LiDAR measurements such as those produced by GLASS or GEDI, but there is no justification for that (most satellites produce continuous measurements). This is surprising given that the authors used continuous RS-based maps to validate such RS sampling strategy, which look like a bit skizophrenic.

Thank you for this important comment. After some critical reflection, we decided to call this method 'transect sampling' as we focus mainly on the establishment of empirical forest plots with this sampling method (lines 111-120). We emphasized the relevance of this sampling strategy also for remote sensing applications in the conclusions section (lines 316-319), as this transect sampling could be interpreted as proxy for airplane flight tracks from lidar campaigns.

Second, I did not fully understand the methodology. I understood that measurements were simulated at different distance along simulated transects but I did not understand how and if the distance between transects varied or not. I am not even sure that the authors simultaneously simulated several transects as would typically be done by a satelitte. I would suggest to simulate a

sampling design similar to the one C2 that was or is adopted by GLASS or GEDI to make this simulation more practical even if this is challenging due to the high resolution of LiDAR footprint (~70 and 20 m) and the abovementioned downscaling problem.

As mentioned above, this transect method sampled plots in North-South transects. Within one transect, the plots had regular distances of 0.5km, 1km or 5 km. The spacing between transects was not regular, but randomly chosen. We revised the method section (lines 112-115) to clarify this approach.

The sampling showed in Fig. 2 illustrates a major problem. Nobody sample at the same time dense humid and dry forests to depict a mean biomass. This is always practically done by forest type using a prior stratification design. The minimum, to have something comparable with the other scales (BCI and Panama) is to focus only on tropical dense humid forests. This may explain why an aggregated sampling design produce such huge errors given that it sample very different forests at the continental scale.

Thank you, this is a good point. In the revised manuscript we combine the biomass map with a biome map (Dinerstein et al., 2017) to distinguish between different vegetation types (see Methods lines 87-90 and revised figure 2). This gives us the possibility to analyze the sampling strategies not only exclusively for tropical forest (covering moist broadleaf, dry broadleaf, conifer and mangrove forest) but also for forests of different biomes separately. After merging these two maps, we found that the number of sampling plots decreases if taking only tropical forest into account (from 102 to 74 for random samples, see revised table 1, "South America (500)"; for aggregated sampling designs see revised figure 6 and revised supplemental figure S4). The additional analysis indicated furthermore, that a stratification into biomes does not lead to a significant reduction of the needed sample plots, since the sum of the plots needed for single biomes (32,200 plots) is similar to the plots needed for an overall forest sampling (36,000 plots). However, this stratification helps to better evaluate sampling effort for each biome. We added the new results (revised figure 4, 5, 6, and S4) and discussed them in the revised manuscript (lines 224-228).

As illustrated in Fig. S3, and by previous studies conducted in BCI, the spatial distribution in AGB do not significantly differs from a random distribution. This explain why, for a given sampled area, using several small or few large plots little impacts your estimates. This should be better explained in the present paper by explicitly mentioning the effect of spatial aggregation on sampling design and by stating that your result would probably not hold at the same scale in many (!) other forests that show strong AGB aggregation patterns (which is the case of most forests).

Thank you for your comment. We will revised the discussion (lines 296-298) by mentioning the effect of spatial aggregation (e.g., Chave et al., (2003); Marvin et al., (2014)).

Note also that the central limit theorem only applies if observations are independents (i.e., in absence of significant spatial structure), such that this theorem is theoretically valid only for the BCI scale in your study.

Thank you for that note, this is a tricky point. It is true that the large scale biomass maps show spatial autocorrelation, but we choose in our sampling strategy "random sampling" the biomass values from random locations of the map. This secures that each of our biomass observation is

randomly generated in a way that does not depend on the values of the other biomass observations. We would have an autocorrelation problem with the used biomass map if we apply for example clustered sampling (i.e., select always nearby points with higher probability). In our study we compared the outcome of the central limit theory only with the random sampling, not with the other sampling strategies. For all other sampling strategies the spatial autocorrelation of the underlying biomass map is crucial – therefore we simulated these sampling strategies instead of applying the central limit theorem. We clarified this in the revised manuscript (lines 108-110).

The discussion section may discuss the realism of a random sampling design at the continental scale in Amazonia.

We discuss the realism of sampling in the revised conclusions section. (lines 312-315)

The conclusion section should highlight more the originality of the present work.

We revised the abstract (lines 20-21) and emphasize transferability of the presented methods in the conclusions (lines 318-320).

**Specific comments**

Line 27: space lacking: "important(Broich" Done.

Line 29: Are those referenced all provided biomass maps? In this study, we compared the biomass maps of Baccini et al., (2012) and Saatchi et al., (2011), but there are more biomass maps available. We added another reference and placed "e.g." at the beginning of the references (lines 30-31).

Line 34: Please replace by "so that the local distribution in biomass". At least remove C3 "local regions", which is inapropriate. Done.

Lines 34-35: This last sentence is very vague. We deleted this sentence.

Line 45: This is an old reference, what about most recent works such as Baccini and Saatchi maps? We deleted this example and cited a more recent example on uncertainty using field plots (lines 48-49)

Line 49: Assume that plots or biomass are. . .. Done.

Line 52: I don't see the logic here. First it is obvious that the representativness of a given number of plots is context-dependent and varies with the total area of interest and second the number of plots fall into the recommendation cited line 48 so that it does not illustrate that the number of plots varies according to the sampling design. We agree that this number does not reflect the differences between sampling strategies. We deleted this example.

Figure 1: I would have personally not call the b panel a landscape scale but rather a regional scale. I know that the definition of scale strongly varies in the litterature but I can hardly imagine a landscape of more than 500 km. Thank you. We renamed the term.

Line 73: "determined using allometric relationship" is really vague, unless the methodology is fully described in the Knapp paper. If yes, please add (see Knapp. . .. For details). The methodology is described in the Knapp paper. We added the reference as proposed.

Lines 77-78: The following sentence is useless and confusing (strange to refer to plots for RS maps, we usually use pixels instead): "For this purpose, between 4 and 25 plots from the original map were averaged." We deleted this sentence and changed the wording from "plot" to "pixels".

Lines 79-80: This last sentence is useless. Deleted.

Line 81-82: This is not true that the Baccini map mostly derived from LiDAR measurements. The global methodology used was to callibrate GLASS LiDAR footprints with field data and then to calibrate a MODIS product with the calibrated LiDAR measurements. Thus the final product mostly reflect MODIS data, that are very little sensitive to biomass and highly sensitive to cloud cover (e.g. the large area of lower biomass observed on the western coastal area of central Africa, compared to the central basin, C4 is simply due to cloud cover). Thank you for mentioning this important point. We revised this sentence (lines 85-87).

Line 83: Please provide rounded numbers. We replaced this sentence by more information on the maps of South America, Africa and South East Asia (lines 84-80).

Line 86-87: Please replace plot by pixel. Done

Fig. 3 legend: "below the bar " should be replaced by "above the bar" Deleted.

Lines 143-144: Very obvious and well-known result. We added " ...the expected result..." (line 152)

Lines 154-155: First sentence useless. Deleted.

Lines 180-181: Please reformulate. Deleted.

Lines 230-232: Very obvious. Deleted.

Line 235: If forest types are known a better strategy would be to stratify the sampling by forest types. We revised the sentence (270-272) and discuss stratification as mentioned above.

Lines 255-256: As already shown and discussed by previous works. We added references (lines 291-292).

Line 259: What is a regional scale here? Forest sites comparable to BCI. We changed to term to "local".

Line 267: For a given sampled area, plot size should not. . .. Done.

Line 270-271: This is what is generally done, remote sensing almost always relies on field data. Please be more explicit. We deleted this sentence and added conclusions for remote sensing sampling (lines 316-319).

***#2 Response to : Interactive comment on "An analysis of forest biomass sampling strategies across scales" by Hetzer et al.***

This manuscript assesses the ability of different sampling strategies to characterize the overall mean biomass of tropical forests. Although there have been previous studies looking at this, the multi-scale approach and the point-pattern simulation to replicate the spatial clustering of previous studies add novelty, and mean that there is enough new for this to be a useful contribution. There are a number of issues that need to be addressed, primarily through improved discussion.

Thank you for your very helpful comments. We prepared a revision of our manuscript that follows your recommendations closely. The main changes are:

d)     Restriction of sampling to forest biomes

Following your suggestion we analyzed the sampling strategies for each biome separately (covering moist broadleaf, dry broadleaf, conifer and mangrove forest). Results are added and discussed.

e)     Discussion about the impact of more variation in high biomass values

The current analysis leads to more conservative estimations. We agree that the tested maps have limitations concerning the fine scale variation. Assuming an increased variation in biomass values would lead to a moderate increase in the minimum sample size. We added this aspect in the discussion.

We have added our responses to your comments in blue following each comment.

**Major comments**

I agree with the major points raised by Ref1, and won't elaborate on them more here except to say that it would make more sense to me to restrict the allocation of sampling points to a single biome (i.e. moist tropical forests) and areas with forest cover (i.e. above a given threshold in the Baccini map) to more realistically reflect real sampling efforts.

Thank you for this comment. We extended the study by analyzing the sampling strategies across different biomes. Therefore, the biomass map used for continental analyses (Baccini et al., 2012) is merged now with a global biome map (Dinerstein et al., 2017). To exclude rarely vegetated pixels within biomes, we assume a minimum above ground biomass threshold of 25 t/ha (see Methods, lines 87-90). Current results show that there are differences between biomes regarding the sampling effort (see revised figure 4 and  figure 5 c-d) We include these additional results in the revised manuscript (lines 174-179 and 224-228).

The analysis of clustered sampling strategies implies a very naïve analysis approach to get an overall mean – just taking an average across plots without considering their configuration. To what extent the performance of clustered plot networks at estimating the overall mean can be improved by analyses accounting of climate and soil covariates and/or spatial autocorrelation to account for this oversampling? I would assume that there would be considerable potential to remove the disruptive effect of non-random sampling, and instead move estimates to a point on the random sampling curve equivalent effective sample size of spatially random plots. Thus existing plot

networks, with appropriate analysis, may provide much better estimates of continental mean biomass than implied by this study.

This is a good point. We designed our analysis primarily to explore the effect of spatially clustered vs. random sampling. We therefore agree that existing plot networks, that stratified plots based on additional constraints, may provide better estimates than suggested by a "blind" clustered sampling.

A possible solution to this issue is to expand the pattern reconstruction approach to include additional criteria (ideally those used for selection of the real clustered plot networks, accounting e.g., for climate and soil covariates). If the covariates representing the additional criteria can be mapped in the entire study area, the pattern reconstruction approach can take the additional constraints into account and reject plot configurations that do not agree with these criteria.

However, we believe that such an analysis would be beyond the scope of our current study, but an interesting task for forthcoming studies. We therefore briefly discuss in the discussion section that our clustered sampling strategies do not account for additional criteria that will be used for the design of real plot networks, and propose the above solution for a better assessment of the performance of clustered plot networks (lines 299-304).

It is worth noting that the remote sensing maps used as reference have serious limitations (some pointed out by Ref1). Most importantly, they miss the effect of species composition on biomass, which is driven by wood density and leads to marked spatial patterns in Amazonia. This isn't so much of a problem for this study if the remote sensing reference maps are interpreted as providing realistic examples of large-scale spatial variation in biomass, rather than as real references. I do wonder if this means the large scale reference maps underestimate the extent of fine scale variation due to compositional differences across soil types (for example).

Thank you for your comment. We agree that continental biomass maps have their limitations in terms of fine scale variation. A higher variation of the biomass variation leads to a higher sampling effort, such that our estimated plot number could be interpreted as a conservative estimation. We added an additional analysis where we assume higher variations in high biomass values (see section S3 in the revised Supplements) and discuss this important issue in the revised manuscript (lines 183-186 and 278-283).

**Specific comment**

The barplots in Figures 3 and 4 could be misinterpreted as giving strong evidence that big plots are best, as they show the that the smaller plot size the more plots are needed. It would be good to also display the change in the area of sampling needed (as is done in the text and table), as that is more relevant to sampling effort.

Thank you. The revised figure 3 includes shapes that represent the total sampling areas.

**#3 Response to: Interactive comment on "An analysis of forest biomass sampling strategies across scales" by Hetzer et al.**

This manuscript details an interesting and novel approach to estimating forest biomass using a dispersed cluster of forest inventory plots. However, in a way this is a "big data" solution to a problem where the solution does not necessarily consider all the variables necessary to making appropriate and constrained biomass estimates. Forest inventory plots are often chosen based on a wide variety of information including forest, soils, hydrology, topography, climate, etc. and are often not randomly chosen. The classical approach of positioning plots in strategic, representative areas often provides constrained and informed estimates of biomass. That said, now that we have huge amounts of remotely sensed data we can apply "big data" approaches to test the extents and limits of many ecological methodologies. I think that is the advantage of this manuscript that it explores this space and does so in an interesting and informative way.

Thank your helpful comments. We prepared a revision of our manuscript that follows your recommendations. The main changes are:

f)      Current forest sampling strategies

We will add paragraphs on current sampling strategies in temperate forests to the introduction. Furthermore, we will discuss the feasibility of adapting those strategies in tropical forests.

We have added our responses to your comments in blue following each comment.

I am honestly torn on whether Amazonia is a perfect test or worst-case scenario fort his methodology. Thinking through this, diversity is incredibly high and there are subset forest types within Amazonia. . ..some that rely heavily on topography/climate such as cloud forests, while there are also dry forests, seasonally flooded forests, and also wet forests. But again, this wide variability may actually be a strength of this approach.

We investigated also the sampling strategies for other continents (see Supplements, Table S1). Results showed that the needed number of randomly sampled plots are higher for the tropical forests of South East Asia (103 plots) and Africa (88 plots) indicating that the tropical forests of Amazonia (74 plots) might be not the worst case, but the best-case-scenario.

I would like to see if compared to temperate forests regions in N. America and Europe as well as boreal areas across the higher latitudes. It would be an interesting comparison to see if those systems diverge wildly from Amazonia.

Thank you for raising this point. Comparisons with temperate regions would be very interesting even though we believe that this would go beyond the scope of this study. For higher latitudes we expect differences in the overall biomass distribution (e.g., due to a different species pool compared to the tropics) but also in the spatial distribution of biomass (e.g., due to forest management). Therefore, there might be also relevant differences in the amount of forest sampling plots needed.

However, forests in temperate regions are mostly monitored with already sophisticated sampling methods (i.e., national forest inventories in North America and Europe).  In the revised manuscript,

we added text about state of the art sampling methods for temperate forests (lines 39-44) and discuss why these sampling approaches are more difficult to establish in tropical regions (lines 312-315).

My specific comments follow, but I think this paper has a lot of potential to drive how we think through sampling and forest inventory methodology. I applaud the creativity of the researchers.

Thank you very much. We appreciate.

Numbers indicates line nos.

14-16: 25 ha is a lot of forest to inventory. I am already thinking of the sheer amount of folks I have to hire. We agree.

26: Define vegetation specifically as aboveground biomass Done.

28: Qualify aboveground Done.

45: These units seem wrong. Also, this is an older citation and only one citation given for what you indicate is a widely varying range. We deleted this example and cited a more recent example on uncertainty using field plots (lines 48-49).

30-55: In general good content, but the case needs to be made why uncertainties in rain forests are potentially higher than other forests. That would add to this section. We added a section in which we compare sampling strategies of tropical forests with those from temperate forest as (see Line 39-44).

Figure 1 : Maybe flip the scale. Done.

Figure 3: What do you mean by accurate here? The caption of figures 3-6 were revised to clarify the term.

* A point. . ..many inventory plots on the ground are circular, but I don't see specifically  (unless I have missed it) but are you using circular or square estimation here? There are some deep literature that may be consulted here about the differences Lindsey, A. A., Barton Jr, J. D., & Miles, S. R. (1958). Field efficiencies of forest sampling methods. Ecology, 428-444.

We are using square plots as we are limited by the spatial resolution of the biomass map. We added this point to the methods (lines 97-99).

228-232: These section could be revisited to ensure clarity in how the results are framed.  Thank you for this comment. We have revised this section (lines 262-264) and added information on the study of Marvin et al.  (2014).

234 – What do you mean by sampling effort increase w/ smaller sample size? We paraphrased this sentence to clarify the statement. (lines 265-267).

**Literature**

[revised manuscript text omitted]

For the first downscaling strategy (D1) we transferred the derived relationships to the South America map. After creating classes over the AGB of 1 t/ha, each AGB value of 25 ha plot of the South American map was assigned to 25 plots of 1 ha drawing random values from a normal distribution $N(AGB_{500}, (sd_{100})^2)$. If the South American plot had an AGB value higher than the maximum value of Panama, the created plots were drawn from a normal distribution with the standard deviation of the maximum class. Negative biomass plots were set to zero.

We analyzed also a second downscaling strategy (D2). Here we assume that the variation of subplots increases linear with biomass (compare Figure S1). For this down-scaling strategy the linear trend resulting from AGB values smaller than 100 t/ha is continued for larger biomass values. Each AGB value of the South American map (1 pixel = 25ha) was assigned to 25 plots of 1 ha drawing random values from a normal distribution $N(AGB_{500}, (m*AGB_{500}+t)^2)$ with slope $m$ and intercept $t$ as coefficients of the linear regression.

[Figure]

(a)

[Figure]

(b)

(c)

**Figure S2** Comparison of downscaling approaches. **a)** Subplot heterogeneity in the Panama biomass map (500 m resolution). Shown is the aboveground biomass (AGB) of plots at a 500 m resolution (x-axis) and the standard deviation (SD) of the associated 25 subplots at a 100 m resolution. (y-axis). Each dot represents one plot from the Panama map (~300,000 plots).The green line represents the downscaling approach D1, as it was implemented in the current study (Table 1). The second downscaling approach D2 shown in pink is based on an increasing linear relationship. **b-c)** Aboveground biomass

 distribution of South America at a 100 m resolution for the two analysed downscaling approaches. Coefficient of variation (CV) and the minimum sample size ($n_{min}$) of randomly chosen 1 ha plots are shown at the upper right corner for each biomass distribution.

**S4 Spatial clustering of the biomass**

[revised manuscript text omitted]

ecology, Ecography (Cop.)., 36(1), 092–103, doi:10.1111/j.1600-0587.2012.07361.x, 2013.

---

## Author Response (AR2)

The authors have taken my comments (and those of the other reviewers) into account and made changes that have improved the manuscript. In particular, I think the new biome-level analysis is a useful addition. The authors have added additional discussion in response to my major comments. This has improved the manuscript, although I think some more consideration is needed of two points.

Thank you for your comments on our revised manuscript. We appreciate the opportunity to resubmit our manuscript with revisions of the points you mentioned below. We have added our responses in blue.

(1) Using the mean of a sample to scale-up to the continental mean. I am happy to concede that an analysis exploring how accounting for climate and soil covariates affects estimates of mean biomass is beyond the scope of the study, and it is good to see there is now some discussion of this issue. However, I am concerned that this will be missed by many readers and this paper will be used to argue that existing plot networks can't be used to obtain the continental-level mean biomass, when it actually shows that plot networks can't capture the overall mean if the non-random nature of sampling isn't accounted for in the analysis (i.e. the comparison of sample mean and true mean (lines 133-140) is a fine way of assessing the relative performance of the sampling methods, but is a much simpler analysis than anyone trying to scale up from plot data would do in reality).
To make sure this important point is clear to the reader, I suggest highlighting it in the abstract. Change line 21 to "Results implied that current plots networks can have clustered structures that reduce the accuracy of largescale estimates of forest biomass if a simple mean is taken of the data". And then maybe add "and ensure clustered sampling is accounted for in analysis of inventory plot data" to line 23. This also improves the practical use of this work – we are unlikely to be able to set up a randomly distributed inventory plot network (although it may be possible to select the locations of some new plots to reduce clustering), but we can make sure we analyse the data from the plot networks to account for the spatial configuration of plots.

Thank you for raising up this point. Following you recommendations closely, we revised the abstract in line 22 adding "…if no further statistical approach is applied" and in line 23 adding "…and ensure that the analyses of inventory plot data consider their spatial characteristics".

I wonder how much the poorer performance of clustered plot networks relates to the detrimental effect of oversampling certain areas versus the failure to adequately sample some areas. If the former is important, then could the point-pattern approach used here to generate spatial patterns that emulate plot networks be reversed to help subsample clustered plot networks to remove the effect of oversampling. If the authors agree this could be important, then perhaps it could be mentioned in the new text (lines 299-304)?

We agree with you. We added a section discussing the opportunities of point pattern analysis (Line 299-304).

(2)Input biomass maps. The additional consideration of fine-scale variation missed by biomass maps is useful. However, I think the authors missed my point about there being large discrepancies between some biomass maps (including the Baccini map they use) and spatial patterns revealed by field data (see Mitchard et al. 2014 Global Ecology and Biogeography). I'm not sure what effect this would have on the analysis, but I think this issue should be briefly discussed.

Thank you. We have constructed a biomass map with 100m resolution, which includes more variability in biomass values which is comparable to the variability found in field data. Analyzing this map, leads to slightly higher estimates of the number of plots needed. We have now made this text clearer (Lines

277-284).

Detailed comments (abstract only)

Line 19 – Odd wording, suggest remove "Though" Done.

Line 20 – Change to "also applied" Done.

Line 24 – I'd remove "This should be considered" from the end of the abstract as it isn't really necessary and is a bit unclear what it is referring to. Done.

[revised manuscript text omitted]